# Applying the Geostatistical Eigenvector Spatial Filter Approach into Regularized Regression for Improving Prediction Accuracy for Mass Appraisal

Michael McCord [1,*], Daniel Lo [1], Peadar Davis [1], John McCord [1], Luc Hermans [2] and Paul Bidanset [3]

1   School of Architecture and the Built Environment, Ulster University, Newtownabbey BT37 0QB, UK
2   Netherlands Council for Real Estate Assessment, Waarderingskamer, 93 210 The Hague, The Netherlands
3   Center for Appraisal Research and Technology, Washington, DC 20009, USA
*   Correspondence: mj.mccord@ulster.ac.uk

**Abstract:** Prediction accuracy for mass appraisal purposes has evolved substantially over the last few decades, facilitated by the evolution in big data, data availability and open source software. Accompanying these advances, newer forms of geo-spatial approaches and machine learning (ML) algorithms have been shown to help improve house price prediction and mass appraisal assessment. Nonetheless, the adoption a of ML within mass appraisal has been protracted and subject to scrutiny by assessment jurisdictions due to their failure to account for spatial autocorrelation and limited practicality in terms of value significant estimates needed for tribunal defense and explainability. Existing research comparing traditional regression approaches has tended to examine unsupervised ML methods such as Random Forest (RF) models which remain more esoteric and less transparent in producing value significant estimates necessary for mass appraisal explainability and defense. Therefore, the purpose of this study is to apply the supervised Regularized regression technique which offers a more transparent alternative, and integrate this with a more nuanced geo-statistical technique, the Eigenvector Spatial Filter (ESF) approach, to more accurately account for spatial autocorrelation and enhance prediction accuracy whilst improving explainability needed for mass appraisal exercises. By undertaking such an approach, the research demonstrates the application of this method can be easily adopted for property tax jurisdictions in a framework which is more interpretable, transparent and useable within mass appraisal given its simple and appealing approach. The findings reveal that the integration of the ESFs improves model explainability, prediction accuracy and spatial residual error compared to baseline classical regression and Elastic-net regularized regression architectures, whilst offering the necessary 'front-facing' and flexible structure for in-sample and out-of-sample assessment needed by the assessment community for valuing the unsold housing stock. In terms of policy and practice, the study demonstrates some important considerations for mass appraisal tax assessment and for the improvement of taxation assessment and the alleviation of horizontal and vertical inequity.

**Keywords:** eigenvector spatial filtering; penalized machine learning; mass appraisal; prediction accuracy; elastic-net regularized regression

## 1. Introduction

There has been increasing emphasis placed on the accuracy of house price and mass appraisal estimation and its role in informing urban, housing and taxation policy. In light of this importance, the accuracy, stability and defenceability of house price models has been a key cornerstone for improving mass appraisal valuation models [1,2]. Typically, the hedonic model has been applied within real estate economics to estimate house prices, which observes house prices to be a function of the property characteristics [3]. However, it is well known that house prices tend to be spatially dependent due to similar physical characteristics shared by neighboring houses and commonalities attributable to

their neighbourhood environment such as access to public facilities and socioeconomic status [4,5]. This spatial autocorrelation (SA), or heterogeneity, has been shown to violate the independent observations assumption required for standard hedonic modelling due to error-bias when not recognizing the spatial heterogeneity of pricing effects [5,6]. This results in the inflation of Type *I* errors which can overestimate the degrees of freedom, reduce confidence intervals and produce bias and inconsistent parameter estimates, leading to inappropriate conclusions [7–9].

This heightened awareness has generated considerable attention for accounting for spatial non-stationarity within house price studies and for mass appraisal estimation. Indeed, insights into the 'confounding effects of space' has been well documented and accounted for in more recent statistical specifications [8,10–12], largely due to advances in spatial information, data sources and both parametric and non-parametric statistical methods. These well-known (geo)statistical methodologies incorporating spatial structural instability into hedonic price modelling have evolved over time which all make use of the spatial characteristics of variables to improve results through reduced error terms and spatial independence [1,13]. Despite these appealing methodological improvements, there remains criticisms as to the stability and superiority of some of these methods, all of which have been subject to scrutiny [6,14–17].

In a similar vein, there has been a research into machine learning (ML) which has gained traction within mass appraisal studies, with a suite of ML algorithms honed and developed since the early 1990s, particularly with respect to its role in Automated Valuation Modelling (AVM) for mass appraisal systems [18,19]. However, these early forms of ML generated some debate with respect to their predictive capacity [20–22] and wider adoption as a consequence of their initial "black box" data-driven nature [10,23] culminating in reduced transparency and opaqueness, both of which are fundamental for defensibility and explainability, particularly within mass appraisal [24,25]. More recent ML approaches have become more prominent due to the increasing availability of open source software packages, codes, digitization and the ability to unearth new pattern recognition which have shown better out-of-sample predictions and valuation accuracy [26–33]. Equally, the "black box" aspect of ML has become less opaque with the augmentation and visibility of (normalized) importance weightings which provide a basis for understanding value significant effects [5].

Yet despite these sizeable improvements, applications of ML have seldom been adopted for mass appraisal modelling or considered SA within their frameworks due to their complexity and unsuitability for public scrutiny [34–37]. Notably, for mass appraisal accuracy and the alleviation of horizontal and vertical inequity, as Sinha et al. [38] contend, if the presence of SA that is not appropriately accounted for or detected, this will affect the training set which will inherently impact upon the test (out-of-sample) data robustness, reliability and prediction accuracy.

This integration of techniques whilst in its infancy has begun to emerge with some contemporary studies [5,37,39–43] starting to extend various algorithms by integrating geo-statistical methods to enhance prediction accuracy, whilst controlling for Spatial dependency. However, to date, fewer studies have explored the application of Eigenvector Spatial Filters (ESF) within supervised forms of Penalized regression algorithms to account for spatial effects for mass appraisal purposes to investigate whether this provides a more suitable methodology for AVM practice and the reduction of inequity.

Therefore the purpose and objective of this study is to apply a methodological approach which provides a simple alternative for including location within traditional regression and supervised Penalized regression approaches while offering a potentially more readily usable approach for property tax assessment jurisdictions [44]. In doing so, this study extends mass appraisal modeling by incorporating eigenvectors generated from a contiguity-based spatial weights matrix to capture unexplained SA within Regularized regression to investigate whether this enhances prediction accuracy in a more adoptable and explainable way. The application of Regularized regression is investigated as these

type of 'supervised' models are more 'front-facing' and transparent for mass appraisal explainability, defensibility and accounting for complex functional forms, model complexity and overfitting [45,46]. The estimated models are developed in training samples and using cross-validation applied to predict market values within the validation sample. The modelling outcomes are further subject to different valuation error measures, and discrimination between the models is determined based on their relative performance.

## 2. Background

Studies investigating house price prediction and accuracy applying various approaches have progressed apace since the 1990s, which Wang and Li [47] in a systematic review of house price modelling, classified into three branches: AI-based models, Geographic Information Systems (GIS) based models and mix-based models.

The application of ML within pricing and mass appraisal studies generally witnessed seminal investigations examining Artificial Neural Networks (ANNs), with much debate and mixed findings, particularly when considering their suggested adoption into mass appraisal. The study conducted by Worzala et al. [19] showed the efficacy of ANNs, however highlighted that they lacked transparency, explainability, and repeatability of results. More recent research also exhibited inconclusive findings with some revealing that ANNs performed better showing increased accuracy and lower predictive error, however, various research has exhibited poorer performance relative to geostatistical methods and in relation to transparency [18,24,25].

Other forms of ML such as Tree classification, Boosted Regression Trees (BRT), and Random Forest (RF) methods have also been propagated within house price studies, and in the main have demonstrated model superiority and the reduction of prediction error when compared to MRAs and other approaches [25,48]. Some however, such as Zurada et al. [49] who comparing various regression and AI-based methods, revealed that regression-based methods were superior with homogeneous datasets, with AI approaches superior with less homogeneous data. Appositely, a number of studies have been somewhat critical of ML techniques, indicating that despite their (varied) superiority in prediction accuracy, they are sensitive to the parameters applied which are not consistent, reliant on data quality and richness, and can suffer from repeatability and model stability [50–53]. In contrast, penalized regression has become a more accepted approach for price estimation especially in the context of mass appraisal, as these types of models do not lack the transparency of ML techniques, and unlike some geostatistical methods such as GWR do not suffer from 'overfitting' due to their shrinkage based approach.

The literature within this area is also emerging as to the efficacy of these regression techniques with studies [54–56] showing reliable estimates and comparable prediction accuracy to other ML approaches. In a similar vein, ESF has also emerged as a reliable and effective approach for mitigating SA due to its ability to integrate into traditional regression-based techniques to produce 'mix-based' geostatistical approaches that are considered transparent and understandable [6,57,58].

As acknowledged by Wang and Li [47], an increasing number of studies are beginning to combine various algorithms with geostatistical (spatial) models to better estimate the real estate value. Studies [37,40,42,43] have all successfully adopted geostatistical approaches within ML architecture to provide improved prediction accuracy and spatial dependence relative to the existing ML specifications. Equally, studies have shown the ESF approach to perform comparably with regression and other geostatistical approaches, but also to comprise some advantages in terms of its ability to identify localized spatial patterns, spatial dependency, residual autocorrelation, and less prone to multicollinearity issues [6,59]. Further, studies have integrated the ESF approach within multilevel modelling and the ML Random Forest and Penalized regression such as Least Absolute Shrinkage and Selection Operator (LASSO) approaches in order to capture spatial heterogeneity and unexplained spatial dependency [5,17,41,60–62] with the findings showing the augmentation of ESF into ML to improve model performance.

The existing literature indicates that there are both advantages and disadvantages to the varying classical regression models and geostatistical and ML techniques. For mass appraisal taxation, regression approaches are more widely known and understood which conforms to the ability to defend and explain readily in a tribunal setting. Similarly, spatial approaches whilst more complex, offer the removal of spatial autocorrelation to produce more reliable parameter estimates and improve predictive accuracy. The advancement of ML has demonstrated more superlative prediction accuracy and error minimization, however has revealed problems within some of the model architectures for mass appraisal in relation to transparency, their data hungry nature and repeatability—notably between training and test samples. Consequently, 'mixed-based' models, similar to validation models in valuation practice, have evolved and have revealed that the combination of geostatistical approaches within regression frameworks can help improve prediction accuracy. In the specific context of mass appraisal, these combinations require some attention in order to conform to explainability and transparency. Therefore, the application of ESF can be readily applied and explained within the more classically orientated regularized regression for the assessment community as this provides parameter estimates devoid of opaqueness whilst accounting for SA. With limited insights currently available within existing literature, this study examines the usefulness of this spatial filtering technique within this type of ML approach.

## 3. Materials and Methods

This section provides descriptions of data, the variables applied within the modelling, an overview of the ESF and Penalized regression approaches employed and the tests for model accuracy.

### 3.1. Data and Variables

The analysis was conducted on a sample of transactional sales data for the Belfast housing market area (UK) obtained from the Ulster University House Price Index (UUHPI) over the period Q2, 2021 and Q2, 2022. In total, 3090 transactions were retained after variable cleansing and erroneous data entry. A data merge was undertaken to obtain the *X*, *Y* coordinates based on the property address using ArcGIS to determine absolute location coordinates. For this study we apply a number of property attributes which represent a key endogenous subset of value significant attributes recognized as the main determinants within the house price literature representing extent and utility which are standard for mass appraisal exercises. A description of the property and neighbourhood variables are presented in Table 1 which includes a number of delineated spatial classifications, boundaries and deprivation ranks obtained from census information to control for locational and neighbourhood attributes.

**Table 1.** Descriptions of property and neighbourhood variables included.

| Variable | Description | Type |
|----------|-------------|------|
| *Property level* | | |
| Sales price | Transacted sales price (£) | C |
| Ln sales price | Logarithmic of sales price (£) | C |
| Floor area | Size of the property in $m^2$ | C |
| Beds | Number of bedrooms, i.e., 1 bed = 1; otherwise 0. | B |
| Baths | Number of Bathrooms, i.e., 1 Bath = 1; otherwise 0. | B |
| Year Built | Age of the property [Year property was constructed] | C |
| Type | Type of property, i.e., Detached = 1, otherwise 0 | B |
| Story | Number of story's | B |
| Garage | Size of garage in $m^2$ | C |

**Table 1.** *Cont.*

| Variable | Description | Type |
| --- | --- | --- |
| *Spatial level* | | |
| Ward | Electoral ward the property is located | B |
| Postcode | Postcode area the property in located | B |
| Locale | Area the property is located. i.e., Suburban = 1; otherwise 0. | B |
| MDM | Multiple Deprivation rank at Electoral ward level | B |

NB: C denotes Continuous; B denotes Binary.

Figure 1 portrays the geographic distribution of house prices across the Belfast Housing Market Area (BMA) over 2021. The distribution shows there to be enclaves, or localised submarkets in terms of the pricing structure, with higher house prices evident towards the South of the BMA, in the East and a small pocket in the North of the City. In contrast, low house prices concentrate in the North and display a radial movement within the inner city over to the East, with a band also stretching from North-west to South west. Overall the house prices reveal a non-uniform distribution and heterogeneity.

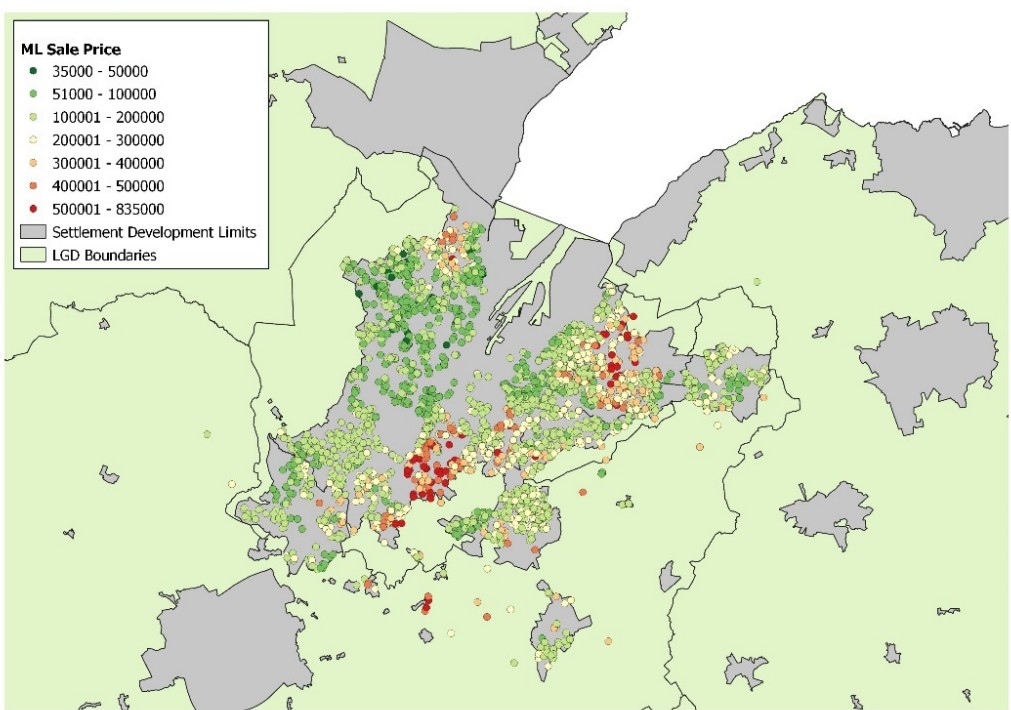

**Figure 1.** The geographic distributions of house prices across the Belfast housing market area.

The descriptive statistics for the property variables can be observed in Table 2. The average sales price over the sample period is £179,163 with the average property size 110 m$^2$. Following the convention of standard ratio study analysis, we adopted a randomly selected set of observations to create a training set (or estimation sample based on approximately 80% of the data and assigned the remaining set of the available data into a testing set (or prediction sample based on 20%).

**Table 2.** Descriptive statistics of the property variables.

| Variable | Mean | Std. Dev | Min | Max |
| --- | --- | --- | --- | --- |
| Sales price | 179,163 | 107,426.517 | 35,000 | 835,000 |
| Ln Sales price | 11.950 | 0.5310 | 10.463 | 13.635 |
| Floor area | 110.226 | 42.801 | 36 | 440 |

**Table 2.** *Cont.*

| Variable | Mean | Std. Dev | Min | Max |
|---|---|---|---|---|
| Beds | 3.032 | 0.786 | 1 | 6 |
| Baths | 1.025 | 0.168 | 1 | 4 |
| Year Built | 1950 | 27.872 | 1810 | 2017 |
| Type | 3.08 | 0.875 | 1 | 4 |
| Build type | 0.84 | 0.371 | 0 | 1 |
| Story | 1.898 | 0.5490 | 1 | 3 |
| Garage | 7.709 | 11.180 | 0 | 119.7 |

### 3.2. Methodology

#### 3.2.1. Machine Learning Regression (Elastic-Net)

There are three well known penalized regression approaches, Ridge, LASSO and Elastic-net, that performs variable selection and regularization both simultaneously. All these types of ML approaches apply a shrinking method by using estimators with smaller variance modifying the cost function in Ordinary Least Squares (OLS) to penalize additional variables in the model, or complexity. The difference between each approach is how they perform their $L_1$-$L_2$ regularization (See Hoerl and Kennard, 1976 for a detailed discussion on Ridge regression and Tibshirani 1996 for LASSO regression). Within this study, we concentrate on the newer form of penalized regression proposed by Zou and Hastie [62], the Elastic-net method, which is a hybrid regularization variable selection method that linearly combines the ridge and lasso regression techniques within a more flexible framework, with the regularization parameter allowing to fluctuate. The approach therefore switches the lambda penalty, when zero is selected it applies the LASSO approach, and when the regularization parameter is one, it becomes a Ridge regression model. Under the Elastic-net regression, the regression coefficient of (1), $\beta$ is estimated by:

$$\hat{\beta}_{ElasticNet} = argmin\beta \left[ (y - X\beta)'(y - X\beta) + \lambda \left( \sum_{j=1}^{p} \left[ (1-\alpha)\beta_j^2 + \alpha|\beta_j| \right] \right) \right] \quad (1)$$

where the hyperparameter $\alpha$ controls how much L1- and L2-norm are used. If $\lambda = 0$, there is no penalty term and $\beta_{ElasticNet} = \beta OLS$. If $\alpha = 0$, $\beta_{ElasticNet} = \beta Ridge$, and $\alpha = 1$, $\beta_{ElasticNet} = \beta_{LASSO}$.

We incorporate the ESF into the Elastic-net model via the set of selected eigenvectors to address any potential issues of spatial correlation. The ESF method introduces a set of spatial matrix eigenvectors ($E_j\delta$) into the regression framework to mitigate SA [63] by applying geographical coordinates that are subject to an eigen analysis of geographical distances to establish a set of spatial filters (eigenvectors) expressing the spatial structure of the region at different scales (for a full explanation see [64]). This interaction of eigenvalues and spatially systematic covariates culminates in eigenvector decomposition which extracts orthogonal and uncorrelated numerical components from the given contiguity matrix [65]. Eigenvectors can be extracted from a doubly centered spatial weights matrix C, expressed as:

$$MCM = \left( I - \frac{11^T}{n} \right) C \left( I - \frac{11^T}{n} \right), \quad (2)$$

where I is an *n x n* identity matrix, 1 is an n × 1 vector of ones, *n* is the number of areal units, T the matrix transpose operator (For a full methodological overview see Griffith (2003)). The set of eigenvectors of MCM, E*full* = {e1, . . . , eN}, provides all the possible distinct map pattern descriptions of latent spatial dependence, with each magnitude being indexed by its corresponding eigenvalue [65]. As discussed by Chun et al. [65], this subset can be identified from a candidate eigenvector set with a stepwise regression procedure (Griffith (2008: 2761) further extended the basic linear model rather than using the final EVs to correct for spatial autocorrelation (SAC) on a global level. Interaction terms are introduced between the selected eigenvectors and the predictors to model spatially varying

coefficients. See Griffith 2008 for a full methodological discussion). The ESF has been applied extensively within a range of modelling techniques. This study further extends the Regularized Elastic-net model by applying ESF and integrating spatial eigenvectors to enhance model prediction accuracy and explanation. In total, five models are developed with a baseline OLS model, an adjusted OLS including the ESF, and standard Elastic-net and an Elastic-net incorporating ESFs.

As observed in Figure 2, the house price data is positively skewed (S: 1.909), we therefore transform the sales price data using the logarithmic to normalize the house price data (S = 0.248). The transformed house price variable is used as the dependent variable across the resulting semi-logarithmic model specifications.

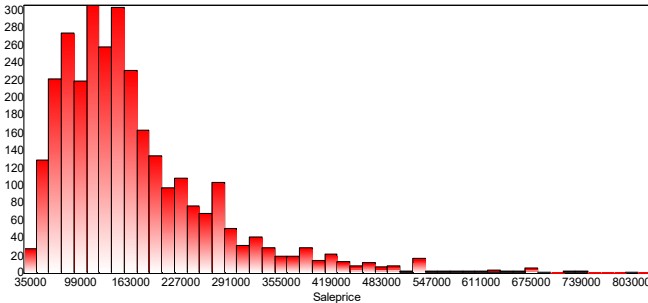

(**a**) house price data

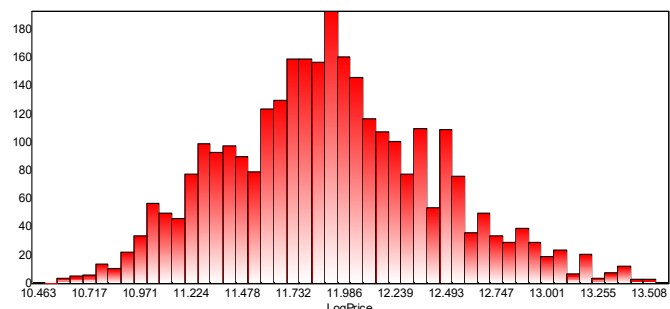

(**b**) Normalized house price data

**Figure 2.** Distribution of sales price across the Belfast Housing Market Area in 2021.

3.2.2. Model Accuracy

To test model performance, the data set is dissected into a training set (in-sample) comprising 80 percent of sales transactions, and a test set (hold-out) composing 20 percent of the sample sales data. The predictive accuracy is measured using three standard measures: the Root Mean Square Error (RMSE), Mean Absolute Error (MAE), and the Mean Absolute Percentage Error (MAPE). International Association of Assessment Officers (IAAO) benchmarks, the Price-Related Differential (PRD) and the Coefficient of Dispersion (COD), are also examined to measure model accuracy for valuation uniformity and inequity.

The RMSE can be defined as the standard sample deviation between the predicted and observed values, with lower RMSE values denoting a better fit model. The RMSE is as follows:

$$\text{RMSE} = \sqrt{\frac{1}{n} \sum_{i=1}^{n} (y_i - \hat{y}_i)^2} \tag{3}$$

where $y_i$ stands for actual and $\hat{y}_i$ stands for the predicted.

The MAE measures the prediction error by taking the mean of all absolute values of all errors. A MAE closer to zero means that the model predicts with lower error and its predictive capacity is superior. The MAE is expressed as:

$$\text{MAE} = \frac{\sum_{i=0}^{n} |y_i - \hat{y}_i|}{n} \tag{4}$$

where $n$ is the number of samples, $y_i$ is the target values, and $\hat{y}_i$ is the predicted values.

The MAPE measures the absolute percentage error in the prediction and can be defined as:

$$\text{MAPE} = \frac{100}{n} \sum_{i=1}^{n} \left| \frac{y_i \hat{-} y_i}{y_i} \right| \tag{5}$$

where $\hat{y}_i$ and $y_i$ stand for the predicted and actual values respectively, while n is the total number of out-of-sample observations.

The COD, is the percentage the average deviation of the ratios from the median, and the most widely used measure of appraisal uniformity. This relative dispersion or

variability of assessments from the median for improved residential property should be <15% and is as follows:

$$\text{COD} = \frac{\frac{100}{n} \sum_{i=1}^{n} \left| R_i - \widetilde{R} \right|}{\widetilde{R}} \tag{6}$$

where $R_i$ is the observed assessment ratio for each parcel, $\widetilde{R}$ the median assessment ratio, and $n$ the number of properties sampled.

The PRD is a mean (valuation to selling price) ratio divided by the weighted mean ratio, which measures the regressivity or progressivity of the assessments. Regressive appraisals occur when high-value properties are under-appraised relative to low-value properties. Progressivity occurs when the opposite happens. If no bias exists, the PRD equals 1, indicating assessment neutrality. Regressivity arises when the values are greater than 1.03; progressivity occurs when values are less than 0.98. The PRD is expressed as:

$$\text{PRD} = \frac{\overline{R}}{\overline{A/S}} = \frac{\overline{R}}{WM} \tag{7}$$

### 3.3. Eigenvector Filter Identification and Explanation

The eigenvectors were created applying a maximum distance connectivity estimation (A number of connectivity criterion algorithms were investigated: the distance criterion $(0 < d > 1)$, the minimum spanning tree, relative neighbourhood, Gabriel criterion, and Delaunay triangulation connections to determine the sampling units), with the spatial filter selection determined by a pre-selection criteria (Threshold set at $p < 0.05$), as the number of filters appointed tends to increase with both level of linear regression residual spatial autocorrelation and the number of areal units. The spatial filters are subsequently examined with the extraction of the filters to be utilised in the regression modelling undertaken using a filter selection criteria with minimisation of the residuals is achieved based on a local Moran's *I* statistic. Overall, 301 (Note that only six spatial filters are demonstrated) spatial eigenvectors filters were determined with the filter selection process applying the Akaike Information Criterion corrected (AICc) and $R^2$ improvement employed to retain those spatial filters where it reduced the AICc statistic. This step therefore minimises the residual short-distance spatial autocorrelation and reduces the level of residual autocorrelation, ensuring model optimality and model stability whilst further encompassing the assessment of each spatial filters spatial correlogram and the variance of the log-price estimation.

This produced 62 spatial filters to be included as independent predictors within the modelling to mitigate spatial autocorrelation and error bias, with a filters showing a coefficient of determination of 47.7 percent (AICc: 78485.929; $p < 0.001$). Further inspection of Figure 3 shows a sample of the extracted spatial filters. Notably, each filter extracted presents a detailed representation of the spatial patterns which can have a different degree of spatial structure, smoothness and geographically varying relationship with house prices. For example, spatial filters one and two capture the initial pronounced structure of market clustering of the eigenvectors which tend to correlate with the underpinning high/low price clusters observed in Figure 1. Notably the spatial structure becomes more 'localised' when displaying the filters (such as Filters 6 and 12) with smaller eigenvalues culminating in more localized parameter surfaces (for example: Filters 25 and 78) given the reduced truncation distances.

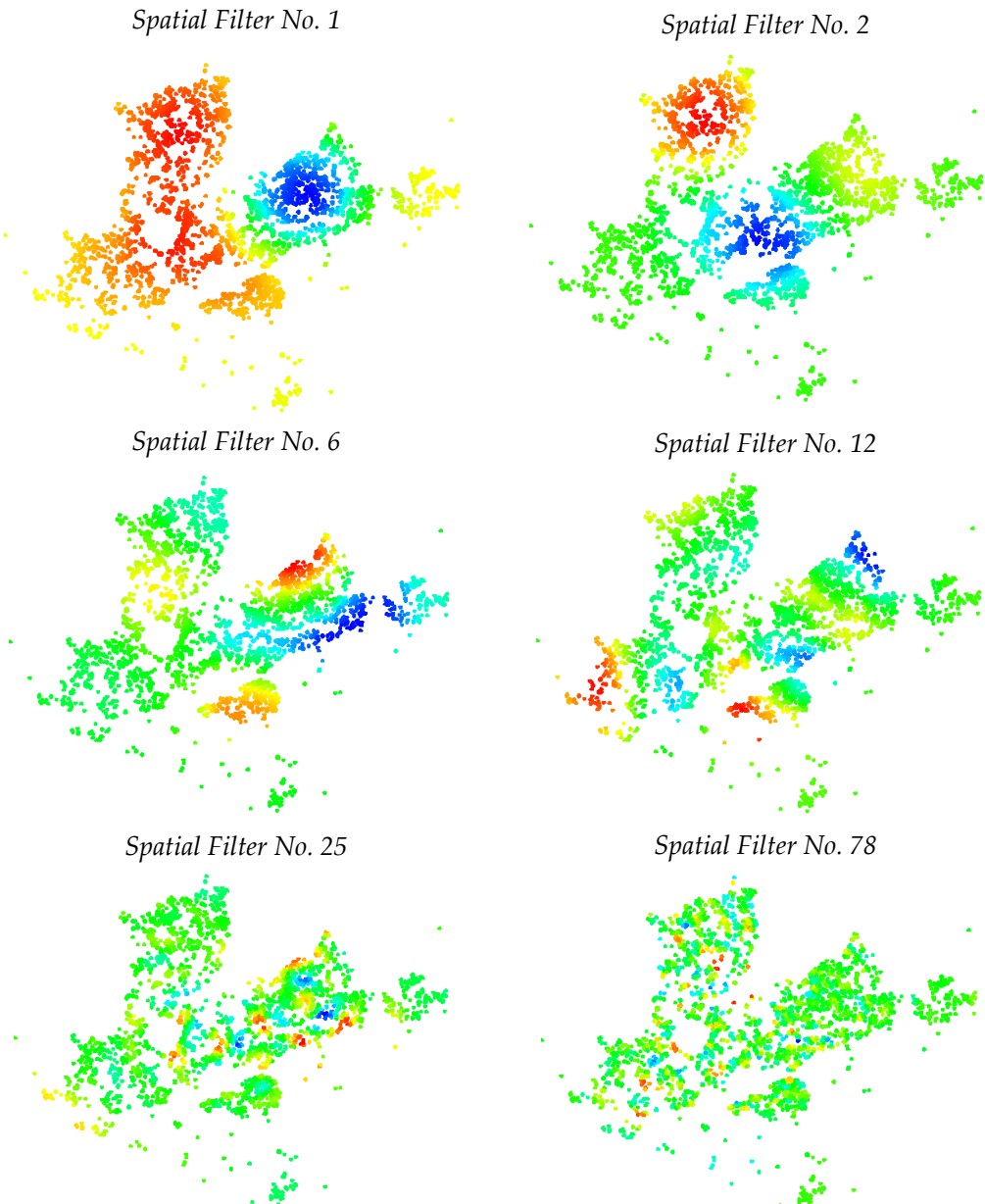

**Figure 3.** Spatial filters extracted across the study area geography.

## 4. Results

The results from the four training and test models are presented in this section. The four training models are specified to account for location information such as the inclusion of delineated boundaries (Models 1 and 3) and the extracted ESFs (Models 2 and 4). Overall, all models exhibit good levels of explanation with model performance ranging between 78.8% and 87.3% (Table 3). The findings reveal the standard Multiple Regression Analysis (MRA) model explains the lowest variation in house prices (78.8%) when incorporating the delineated spatial information. With the inclusion of ESFs, the level of explanation increases by 5.1% demonstrating an Adjusted $R^2$ of 83.8%. The level of explanation further increases when examining the Elastic-net ESF which observes an $R^2$ of 87.3%.

The model coefficients reveal, by-and-large, the expected signs, magnitudes and significance (Within this study, for the penalized regressions we report only the values of $\lambda$-min (lambda)). For the standard OLS and Elastic-net models, apartments show negative coefficients of 9.1%, however, for the augmented ESF models this effect increases to 13.9%. The same observation is notable for the terrace coefficient which also increases in magnitude

by approximately 5.8% between the model specifications. The analysis shows for a unitary increase in one squared metre, price, increases by 0.5%. Similarly, property age (Year built) exhibits a negative coefficient symbolizing that for every one year decrease in property age, price decreases by 0.2%.

**Table 3.** Training models level of explanation and coefficients.

| | Model 1 | Model 2 | Model 3 | Model 4 |
|---|---|---|---|---|
| **Variable** | **MRA** | **Elastic-Net** | **MRA + ESF** | **Elastic-Net + ESF** |
| | Coefficient | Coefficient | Coefficient | Coefficient |
| Apartment | −0.09099 * | −0.09096 | −0.13904 ** | −0.13899 |
| Detached | 0.143561 *** | 0.143531 | 0.154712 *** | 0.154662 |
| Terrace | −0.12268 *** | −0.12265 | −0.18009 *** | −0.18004 |
| Size (m$^2$) | 0.0052 *** | 0.0052 | 0.0050 *** | 0.005 |
| Year Built | −0.0005 *** | −0.0005 | −0.0002 | −0.0002 |
| Bed 4 | 0.0324 * | 0.0323 | 0.0225 | 0.0224 |
| Bed 5 | 0.0012 | 0.0012 | 0.0233 | 0.0232 |
| Bed 6 | 0.0944 * | 0.0944 | 0.068 | 0.0679 |
| Bath 2 | 0.0145 | 0.0144 | 0.0062 | 0.0061 |
| Bath 3 | −0.2777 * | −0.2777 | −0.2740 ** | −0.2739 |
| Garage size (m$^2$) | 0.0025 *** | 0.0025 | 0.0027 *** | 0.0027 |
| Grade B | −0.0358 | −0.0357 | 0.0049 | 0.0049 |
| Grade C | −0.0367 ** | −0.0367 | −0.0489 *** | −0.0488 |
| Story 1 | 0.0547 ** | 0.0547 | 0.0511** | 0.0511 |
| Story 1.5 | −0.0462 | −0.0462 | −0.029 | −0.029 |
| Story 2.5 | 0.0309 | 0.0309 | 0.0238 | 0.0238 |
| Story 3 | 0.0341 | 0.0341 | −0.0318 | −0.0317 |
| Rural | −0.2003 *** | −0.2003 | | |
| Rural village | −0.3152 *** | −0.3152 | | |
| Suburban | −0.0022 | −0.0021 | | |
| MDM | 0.0001 *** | 0.0001 | 0.0001 *** | 0.0001 |
| PC_2/SF_2 | 0.7429 *** | 0.7429 | −4.9570 *** | −4.957 |
| PC_3/SF_3 | 0.1025 *** | 0.1025 | 4.1330 *** | 4.133 |
| PC_4/SF_4 | 0.1000 *** | 0.0999 | 0.5410 * | 0.541 |
| PC_5/SF_5 | 0.1206 *** | 0.1206 | −0.334 | −0.334 |
| PC_6/SF_6 | 0.3131 *** | 0.313 | 0.015 | 0.015 |
| PC_7/SF_7 | 0.0867 *** | 0.0867 | −1.9810 *** | −1.981 |
| PC_8/SF_8 | 0.3303 *** | 0.3303 | −1.7050 *** | −1.705 |
| PC_9/SF_9 | 0.1549 *** | 0.1549 | 2.3320 *** | 2.332 |
| PC_10/SF_10 | 0.1233 *** | 0.1233 | 0.5000 ** | 0.5 |
| PC_11/SF_11 | −0.0717** | −0.0716 | −0.355 | −0.355 |
| PC_12/SF_12 | −0.1976 *** | −0.1975 | 0.7620 *** | 0.762 |
| PC_13/SF_13 | −0.1889 *** | −0.1889 | 1.1500 *** | 1.15 |
| PC_14/SF_14 | −0.0922 *** | −0.0921 | 0.081 | 0.081 |
| PC_15/SF_15 | −0.0301 | −0.0301 | −0.6920 *** | −0.692 |
| 2021Q | −0.0805 *** | −0.0804 | −0.0828 *** | −0.0828 |
| 2022Q | −0.0466 *** | −0.0465 | −0.0446 *** | −0.0446 |
| C | 10.532 | 10.584 | 10.6235 | 10.6855 |
| R-squared | 0.788 | 0.822 | 0.843 | 0.873 |
| Adjusted R-squared | 0.787 | | 0.838 | |
| L1 Norm | | 22.91142 | | 14.699 |
| F-statistic | 397.8216 *** | | 329.7989 *** | |
| N | 2481 | 2481 | 2481 | 2481 |

NB. SF denotes spatial filters. Only the first 15 spatial filters have been presented due to space limitations and for presentation purposes to align with postcodes. The remaining ESFs are available upon request. Penalized models are specified using Regressor transformation: Std Dev (smpl). Cross-validation method: K-Fold (number of folds = 5), rng = kn: seed = 115121966, and the Selection measure: Mean Squared Error. *** denotes sig. 0.001 level; ** 0.05 level; * 0.10 level.

Whilst the findings reveal a notable increase in model performance, the inclusion of the ESFs has reduced the residual error and level of spatial autocorrelation across the study area geography (Figure 4a–d), with the delineated spatial models demonstrating larger clusters of heightened residuals within specific locales. In essence, there appears larger differences between the actual and estimated sales prices which the standard regression and Elastic-net models have not successfully addressed as a consequence of SA and not detecting the underpinning spatial patterns.

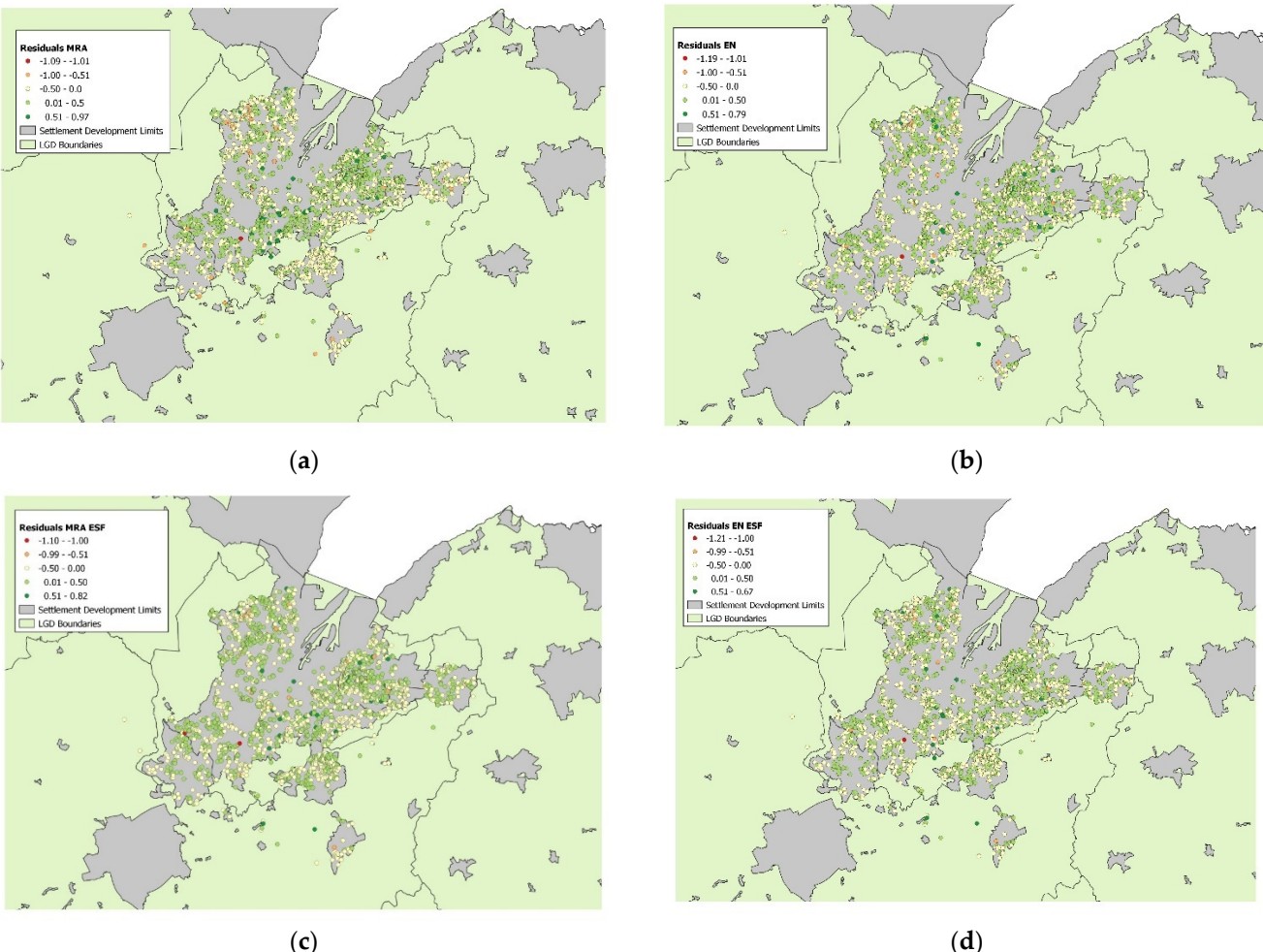

(**a**)   (**b**)

(**c**)   (**d**)

**Figure 4.** Residual errors across the different models. (**a**) MRA model. (**b**) Elastic-net model. (**c**) MRA ESF model. (**d**) Elastic-net ESF model.

*Model Prediction and Accuracy*

The analysis examines the accuracy of the predictive capacity of the training and test models. As observed in Figure 5, the scatterplots display the for the training models the comparisons between the assessed (estimated) values and the observed sales price—the Assessment to sales price ratio. The analysis reveals the standard MRA to display a correlation of 74.1% between the observed and predicted, however the presence of heteroskedasticity is noticeable, particularly at the higher end of the price strata, which is symbolic of mass appraisal regressivity—where higher valued properties are under-appraised relative to lower valued properties. The Elastic-net observed and predicted values shows an increase in the relationship between the assessed and observed sales price with a correlation of 77.2% and a reduction in the level of heteroskedasticity, albeit this is still evident. The ESF regression and Elastic-net models show improvement in their respective correlations (81.2% and 83.6%), some 9.5% increase from the baseline MRA, increased linearity and the reduced presence of heteroskedasticity.

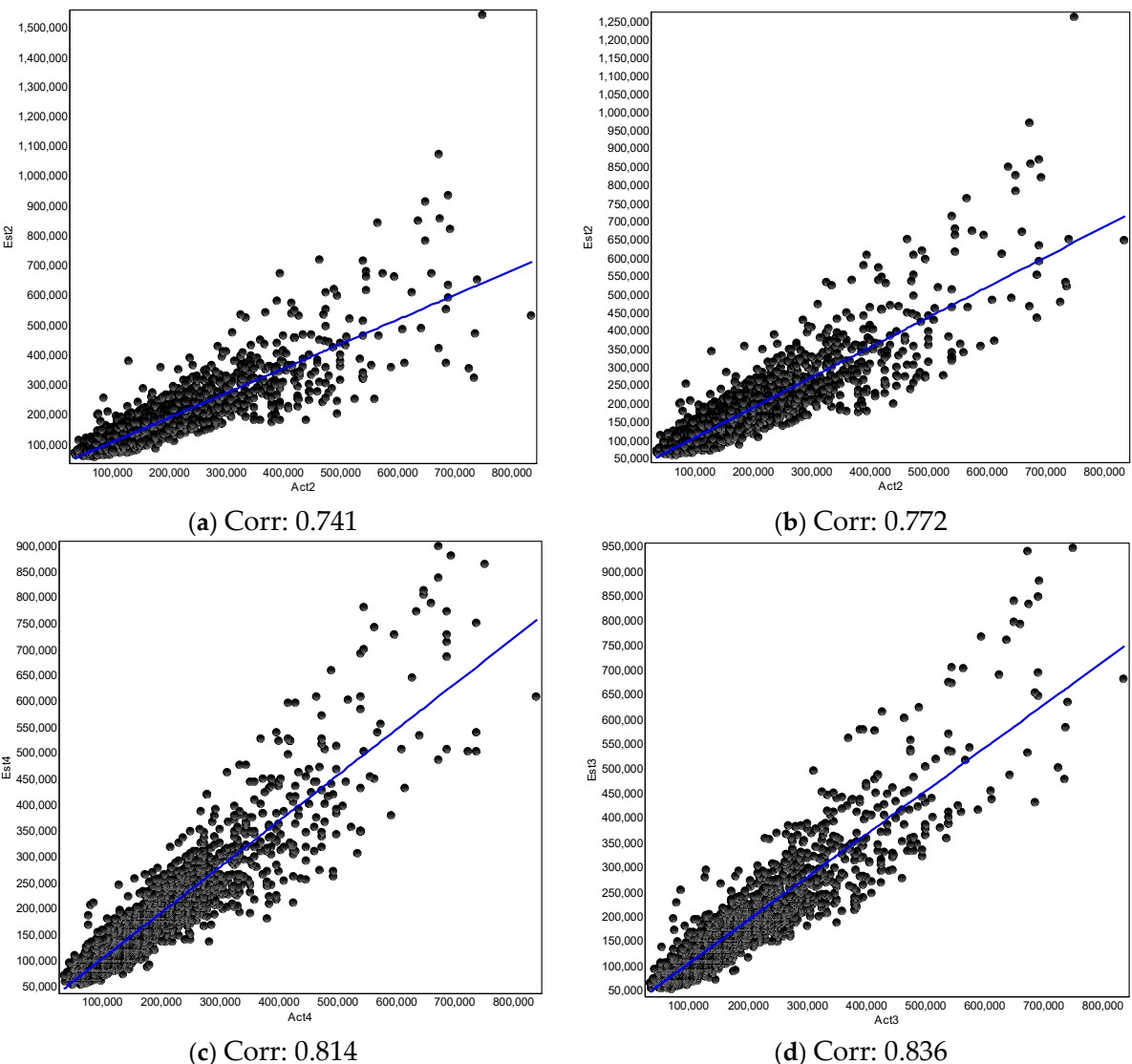

**Figure 5.** Scatterplots of observed house vs. predicted prices. (**a**) MRA model. (**b**) Elastic-net model. (**c**) MRA ESF model. (**d**) Elastic-net ESF model.

Table 4 further provides a summary of the accuracy and ratio statistics for each model. The findings exhibit the standard MRA to show less accuracy than the other models across all metrics analysed with the highest RMSE (22.98%), MAE (18.07%) and MAPE of 1.517 across the training dataset. In comparison, The ESF models show sizeable improvement on the predictive accuracy with the augmented MRA incorporating ESF noting a RSME of 21.08%, MAE of 16.39% and MAPE of 1.378. The Elastic-net ESF model performed the best exhibiting the lowest RSME (20.76%), MAE (15.93%) and MAPE (1.368). This is also evident for the test dataset which also reveals the prediction accuracy to be superior and more accurate for the Elastic-net models than the hedonic counterparts. The models which integrate the ESFs produce the most accurate predictions exhibiting the smallest RSME, MAE and MAPE statistics for the out-of-sample testing (Table 4).

The ratio statistics which measure prediction accuracy by testing for inequity and uniformity using the IAAO benchmarks, reveal the models which comprise the spatial filters to perform best. Examination of the COD for the training set data reveals the MRA to perform worst (19.4%) with the Elastic-net model integrating the spatial filters to perform best (15.9%) and only marginally falling outside the acceptable boundary of assessment uniformity. This is also the case for the PRD with the standard MRA exhibiting regressivity and beyond the accepted boundary of 1.03. Again, and notably, the Elastic-net containing

the ESFs is superior and only slightly beyond the tolerance for appraisal inequity for the training and test data. When considering the out-of-sample testing, the findings show the MRA approach to be the least accurate, displaying heightened levels of assessment inequity and a decrease in uniformity compared to the other approaches. The Elastic-net however does not show as much pronounced differences, invariably due to the shrinkage approach applied to the coefficients through the cross-validation. The MRA based ESF and Elastic-net ESF exhibit the least variance between the in- versus out-of-sample tests.

**Table 4.** Model prediction accuracy and ratio statistics.

| Model | | RMSE | MAE | MAPE | PRD | COD | N |
|---|---|---|---|---|---|---|---|
| MRA | Train | 0.2298 | 0.1807 | 1.5172 | 1.057 | 0.194 | 2841 |
| | Test | 0.2351 | 0.1864 | 1.5659 | 1.074 | 0.211 | 602 |
| Elastic-net | Train | 0.2151 | 0.1787 | 1.5099 | 1.053 | 0.185 | 2841 |
| | Test | 0.2268 | 0.1823 | 1.5358 | 1.061 | 0.190 | 602 |
| MRA + ESF | Train | 0.2108 | 0.1639 | 1.3777 | 1.044 | 0.171 | 2841 |
| | Test | 0.2194 | 0.1687 | 1.3905 | 1.049 | 0.179 | 602 |
| Elastic-net +ESF | Train | 0.2076 | 0.1593 | 1.3682 | 1.032 | 0.159 | 2841 |
| | Test | 0.2134 | 0.1638 | 1.3811 | 1.038 | 0.167 | 602 |

## 5. Discussion

The discussion and testing of model performance has tended to examine superiority comparing the differing geo-statistical, traditional regression and ML approaches, all of which show advantages and disadvantages for each, and different magnitudes of prediction accuracy, which are all very much data and study area dependent. Of late, the scrutiny of these different approaches has centered more on their amalgamation within a mixed or hybrid framework, and specifically how to optimise model specifications to account for the spatial variation of house prices.

For mass appraisal, the alleviation of horizontal and vertical inequity and uniformity is of paramount importance. The results emerging from this study demonstrate that the inclusion of ESFs accounting for SA enhance model explainability and predictive accuracy compared to classic MRA and ML Elastic-net models which use delineated postcode proxies to account for spatial heterogeneity. This finding is in accord with existing research studies employing mixed-based approaches [5,17,41,61] who also found that the inclusion of spatial eigenvectors derived from geographic coordinates improved model performance relative to other ML or regression-based models applying other types of spatial information as proxies. Pertinently, the findings also revealed reduced spatial error and more stable residuals when including spatial filters, again in keeping with extant research [5,17,60,62]. The alleviation of spatial residual error also improves the out-of-sample tests for accuracy, a finding in keeping with Sinha et al. [38] who identified that the failure to adequately mitigate SA reduces model reliability and test accuracy.

This finding is an important issue when undertaking mass appraisal exercises. The in-sample versus out-of-sample performance is of primary concern when using the model to subsequently value the unsold housing stock and for improving both horizontal and vertical inequity and uniformity within mass appraisal assessment. Indeed, as identified in the study of Hu et al. [5], the superior model performance and prediction accuracy resulted from the addition of coordinate variables are likely to be attributable to the well-matched spatial patterns observed in coordinate variables and house sale price data, and the results do from both a visual and inferential perspective show the spatial eigenvectors to mirror market structure, topography and submarkets which leads to model improvement through the capture of spatial patterns or processes. In contrast, the application of delineated boundaries within ML and other classical regression approaches result in not only the confounding issues of SA, but also lack of explanation relative to house prices as they rarely match housing sub-markets, and further open to scrutiny when considering omitted variable bias.

This study has found that the identification and extraction of the spatial filters reduces any potential for this bias to occur as the spatial matrix eigenvectors minimise residual autocorrelation based on the spatial structure of the study area at varying scales, and can be regarded as patterns of independent spatial dimensions, culminating in the almost complete elimination of residual spatial autocorrelation and therefore mitigating parameter estimation bias and helping to account for unexplained spatial patterns. For the field of mass appraisal, this identification of the spatial structure can help more accurately identify local submarket fluctuations, leading to better ratio studies and more uniform, equitable, and accurate valuations which can help save costs associated with inequity. Further the incorporation of ESFs can greatly reduce the amount of time it takes to create multiple sub-models or a flexible global model, making it more efficient for mass appraisal purposes.

Existing research has examined the role of ML within mass appraisal [18,25–28], with early arguments critiquing the 'black-box' nature of the model outputs, and despite improvements in the reporting of 'importance' plots providing information for assessors [35–38] remains challenging for wholesale uptake within mass appraisal practice given the complexity and repeatability of these types of algorithms. This study shows that the application of Regularized regression incorporating spatial filters is a more obvious choice for the assessment community, and for taxpayers, as the ESF approach provides a foundation for including location providing market professionals and policymakers with a more readily and understandable methodology for applying spatial analysis in a more standardised and explainable hedonic framework for understanding housing markets and for applications seeking to harness such understanding, such as automated valuation modelling for mortgage lending, or mass appraisal of residential values for property taxation purposes.

## 6. Conclusions

The role of spatial autocorrelation within house price studies and mass appraisal has been an increasingly important topic for the generation of accurate price and valuation estimations. This has evolved particularly due to the advancement of geo-statistical techniques and ML approaches driven by developments within data, its accessibility and open access software packages. Despite these innovations, until recently, existing ML algorithms and studies have often neglected to account for SA. Recent developments within house price analysis has begun to integrate hybrid or mixed-based approaches to augment explanatory power and account for spatial dependency to improve prediction accuracy. Indeed, an emerging corpus of existing studies have shown the efficacy of this type of integration of spatial eigenvectors. However, despite this promising line of research, ML learning approaches such as ANNs, Random Forests and Decision Trees remain limited in their uptake, particularly for mass appraisal purposes, principally due to a lack of transparency and complexity which is challenging for assessment jurisdictions to defend for public accountability.

The literature has been advancing investigating the incorporation of geo-statistical approaches and more latterly the selection of spatial eigenvectors within machine learning algorithms. Therefore, this paper sought to extend both the standard regression and ML regularized Elastic-net approaches by proposing a new hybrid model incorporating eigenvector spatial filters in order to develop a flexible ML spatial methodology which mitigates SA whilst offering a more readily transparent approach to improves house price prediction and mass appraisal.

The empirical findings emerging from this study contribute to knowledge in three ways. Firstly, the research advances the integration of the ESF geo-statistical approach within ML for mass appraisal purposes.. In doing so it provides insights into developing a more understandable and usable approach for assessment communities which will stand up to tests for defensibility and explainability as opposed to other ML approaches. Thirdly, it establishes that the integration of spatial filters show improved efficacy and predictive capacity on baseline classical regression and Elastic-net architecture by reducing spatial residual error. Indeed, the empirical findings demonstrate the exploratory capacity and

capability of the Elastic-net ESF model for accommodating SA inherent within sales prices and producing a ML model which offers the necessary 'front-facing' technique which is readily implementable and flexible structure which provides enhanced price prediction for in-sample assessment and out-of-sample assessment which is needed by the assessment community for valuing the unsold stock. These analytical insights thereby offer a more user-friendly adaptable approach for enhancing mass appraisal.

In terms of policy and practice, this study has demonstrated some important considerations for mass appraisal tax assessment and for the improvement of taxation and housing policy. Machine Learning has the power to interpret data to provide insights that are not immediately apparent from the available data. Integrating geo-statistical methods not only improves the power and performance of ML, it also provides a coherent structure to interrogate and display the findings in a manner which is more appealing and intelligible to practitioners and policy-makers, all of which operate in a spatial reality of communities and economic landscapes, and who necessitate understanding of how more abstract analysis relates to reality.

Future studies may wish to investigate other ML based applications and the integration of spatial filtering to establish whether comparable performance is achieved when comparing the unsupervised forms of ML with the supervised forms of ML and the robustness and accuracy of each. On a cautionary note, there remains some challenges, namely voluminous datasets and the computational time required to extract the spatial filters due to the large set of interaction terms required and the automation of the more complex computational steps. Further, Finally, it is meritorious to note that whilst the ESF and ML regularization techniques offer assessment jurisdictions and appraisers more opportunity for creating more accurate appraisals, this now requires advanced knowledge of data science, statistical insights, and application which remains a challenge for assessment authorities in practice.

**Author Contributions:** All authors listed meet the authorship criteria and are in agreement with the submission of the manuscript. Conceptualization, M.M., P.D., J.M. and D.L.; methodology, M.M., P.B., L.H. and D.L.; software, M.M., P.D. and L.H.; validation, all authors; formal analysis, all authors; writing—original draft preparation, all authors; writing—review and editing, all authors; visualization, P.B., J.M. and D.L.; project administration, all authors. All authors have read and agreed to the published version of the manuscript.

**Funding:** This research received no external funding.

**Institutional Review Board Statement:** Not applicable.

**Informed Consent Statement:** Not applicable.

**Data Availability Statement:** The data that support the analysis of this study are available from the corresponding author, [MM], upon reasonable request.

**Conflicts of Interest:** The authors declare no conflict of interest.

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
