# Peer review of "Applying the Geostatistical Eigenvector Spatial Filter Approach into Regularized Regression for Improving Prediction Accuracy for Mass Appraisal"

_applsci, doi:10.3390/app122010660_

Round 1

Reviewer 1 Report

This study aims to use a geostatistical approach, namely the Eigenvector Spatial Filter (ESF), into regularized ML to account for spatial autocorrelation and improve prediction accuracy and the explainability of ML models.

The authors claim that the study results show that the integration of spatial filters improves model explainability and prediction accuracy with reduced spatial residual error by removing spatial autocorrelation.

The manuscript has an interesting topic, it is correctly written, quite well organized and essay to read and understand, well documented, but some parts are still fragile.

The authors didn’t highlight the relevance of the study and which brings new, the innovative elements, without these the research is a simple case study. It is not clear from this manuscript what the purpose and necessity of this study is. If these disadvantages are clarified, the manuscript could be published successfully.

My questions are: What is the research question or research gap you would like to answer?

What is novel in this paper? Could you expose it?

The purpose and the objectives of the research should clearly defined, also the relevance of the research and the novelty elements should be presented.

The abstract must contain the purpose of the paper, the methods used in the research and a brief presentation of the results and conclusions.

The introduction should contain the purpose and the objectives of the research clear defined and a brief presentation of the methodology.

I recommend introducing the names of the localities inside the maps, only their limits are not enough.

In table 2, page 6 are presented property variables, the relevance of these variables should be explained, why were they chosen? Why are these the most important?

Author Response

The authors didn’t highlight the relevance of the study and which brings new, the innovative elements, without these the research is a simple case study. It is not clear from this manuscript what the purpose and necessity of this study is. If these disadvantages are clarified, the manuscript could be published successfully.

My questions are: What is the research question or research gap you would like to answer?

What is novel in this paper? Could you expose it?

We have updated the text to reflect the reviewers comments

The purpose and the objectives of the research should clearly defined, also the relevance of the research and the novelty elements should be presented.

The abstract must contain the purpose of the paper, the methods used in the research and a brief presentation of the results and conclusions.

This has been updated to reflect the reviewers comments

The introduction should contain the purpose and the objectives of the research clear defined and a brief presentation of the methodology.

This has been updated to reflect the reviewers comments

I recommend introducing the names of the localities inside the maps, only their limits are not enough.

The maps all contain Local Government District delineated boundaries with the exception of the spatial filters. The filters are presented in this manner to show the eigenvector clustering and heterogeneity and removal of SA.

In table 2, page 6 are presented property variables, the relevance of these variables should be explained, why were they chosen? Why are these the most important?

The text has been updated to explain the relevance and selection of the variables

Reviewer 2 Report

This paper presents a solution for improving the accuracy of regression methods applied to the field of real estate price prediction. The idea is to introduce an eigenvector filter approach, an approach already known to be able to reduce spatial autocorrelations inherent to geographic data.

The article is rather difficult to follow and deserves, in my opinion, important corrections to convince of its contribution.

The introduction and background section are redundant, repeating the same observations about the various existing approaches but the discussion is not structured and rather confusing. Similar arguments are also in section Discussion. A table or some other form of synthesis of the pros and cons presented would be welcome.

Since the presented work involves integrating spatial eigenvector filtering with a regression method and the advantage of this solution has already been highlighted in other works (for instance Zhang J, Li B, Chen Y, Chen M, Fang T, Liu Y. Eigenvector Spatial Filtering Regression Modeling of Ground PM2.5 Concentrations Using Remotely Sensed Data. Int J Environ Res Public Health. 2018), it is important to show the contribution of the paper to the state of the art. Is it the originality in working in the field of real estate price prediction? Or is it in the choice of the Elastic-net regression method? Either one does not seem to me to be a convincing contribution.

The exact purpose of the work presented is unclear. The arguments in the introduction and conclusions are unclear and disjointed.

The argument that "limited knowledge is currently available in the existing literature" does not seem to be sufficient, indeed it is misleading.

In addition,

-       a number of acronyms are not detailed as they should be: OLC, MRA, ACI, IAAO…

-       line 268 it is not a definition of C, and 11 should be explained

-       line 354 what about filters 6 and 12, they are not commented

Author Response

This paper presents a solution for improving the accuracy of regression methods applied to the field of real estate price prediction. The idea is to introduce an eigenvector filter approach, an approach already known to be able to reduce spatial autocorrelations inherent to geographic data.

The article is rather difficult to follow and deserves, in my opinion, important corrections to convince of its contribution.

The introduction and background section are redundant, repeating the same observations about the various existing approaches but the discussion is not structured and rather confusing. Similar arguments are also in section Discussion. A table or some other form of synthesis of the pros and cons presented would be welcome.

We have tried to streamline the paper to make it more readable and easier to follow. We have trimmed down the introduction to reduce repetition and sharpened the focus and add the rationale and purpose. We have attempted to revise aspects of the introduction and removed text which may have been repetitive or confusing. Whilst we have not included a table in the discussion as per the reviewers comments, we have included the advantages for this type of new method for mass appraisal assessment.

Since the presented work involves integrating spatial eigenvector filtering with a regression method and the advantage of this solution has already been highlighted in other works (for instance Zhang J, Li B, Chen Y, Chen M, Fang T, Liu Y. Eigenvector Spatial Filtering Regression Modeling of Ground PM2.5 Concentrations Using Remotely Sensed Data. Int J Environ Res Public Health. 2018), it is important to show the contribution of the paper to the state of the art. 

We acknowledge the reviewers comments. We have attempted to distil out the contribution of knowledge and where it relates to advancing mass appraisal ‘state of the art’ and insights into AI for mass appraisal purposes

Is it the originality in working in the field of real estate price prediction? Or is it in the choice of the Elastic-net regression method? Either one does not seem to me to be a convincing contribution.

The originality of the work rests in the advancement of the integration of a geo-statistical approach within AI for mass appraisal purposes. It provides insights into developing a more understandable and usable approach for assessment communities which will stand up to tests for defensibility and explainability as opposed to other ML approaches. It also establishes that the integration of spatial filters reduced SA and enhance model predictive capacity and reliability in comparison to Elastic-net and regression models which apply delineated locational parameters. We have included this within the text.

The exact purpose of the work presented is unclear. The arguments in the introduction and conclusions are unclear and disjointed.

We have removed/updated the text where applicable to showcase the purpose of the work

The argument that "limited knowledge is currently available in the existing literature" does not seem to be sufficient, indeed it is misleading.

We have revised this statement.

In addition,

-       a number of acronyms are not detailed as they should be: OLC, MRA, ACI, IAAO…

-       line 268 it is not a definition of C, and 11 should be explained

-       line 354 what about filters 6 and 12, they are not commented

These have been revised. All acronyms are in text, the definition of C and 11 are included as follows:

As defined by Griffith and also evident in the work of Hu et al (2019); C is denoted as doubly centered spatial weights matrix, where I is an n x n identity matrix, 1 is an n x 1 vector of ones, n is the number of areal units, T the matrix transpose operator

Both filters have been commented upon in text.

Reviewer 3 Report

It is a very good paper that can change the path for many researchers aimed to find how new trends of technology can combine with old needs such as "mass appraisal". The lack of quality data such as "asking price vs price paid" is still an issue in many countries but the method here is well presented. I think it is ready to be published in the present form.

Author Response

We thank the reviewer for their kind comments.